# Foliar Silicon Alleviates Water Deficit in Cowpea by Enhancing Nutrient Uptake, Proline Accumulation, and Antioxidant Activity

**DOI:** 10.3390/plants14081241

**Published:** 2025-04-19

**Authors:** Larissa Lanay Germano de Queiroz, Evandro Franklin de Mesquita, Caio da Silva Sousa, Rennan Fernandes Pereira, José Paulo Costa Diniz, Alberto Soares de Melo, Rayanne Silva de Alencar, Guilherme Felix Dias, Vitória Carolina da Silva Soares, Francisco de Oliveira Mesquita, José Philippe Martins Montenegro Pires, Samuel Saldanha Rodrigues, Lays Klécia Silva Lins, Anailson de Sousa Alves, Karoline Thays Andrade Araújo, Patrícia da Silva Costa Ferraz

**Affiliations:** 1Graduate Program in Agricultural Sciences, Paraíba State University, Campina Grande 58429-500, PB, Brazil; larissalanay.bio@gmail.com (L.L.G.d.Q.); patriciagroambiental@gmail.com (P.d.S.C.F.); 2Department of Agrarian and Exact, Paraíba State University, Catolé do Rocha 58884-000, PB, Brazil; evandrofranklin@servidor.uepb.edu.br (E.F.d.M.); caiosilvafla16@gmail.com (C.d.S.S.); josepaulo.rc06@gmail.com (J.P.C.D.); vitoria.16carolina@gmail.com (V.C.d.S.S.); philippemp97@gmail.com (J.P.M.M.P.); samuelsaldanha90@gmail.com (S.S.R.); layslins@servidor.uepb.edu.br (L.K.S.L.); anailson.alves@servidor.uepb.edu.br (A.d.S.A.); karoline_thays@hotmail.com (K.T.A.A.); 3Department of Biology, Paraíba State University, Campina Grande 58429-500, PB, Brazil; alberto.melo@servidor.uepb.edu.br (A.S.d.M.); rayannesilvadealencar@gmail.com (R.S.d.A.); guilhermefelix038@gmail.com (G.F.D.); 4Department of Ecology, Paraíba State University, Campina Grande 58429-500, PB, Brazil; mesquitaagro@yahoo.com.br

**Keywords:** abiotic stress, orthosilicic acid, *Vigna unguiculata*, water use efficiency, yield

## Abstract

Silicon has emerged as a beneficial element in mitigating water deficit in various crops, although the underlying mechanisms still require further investigation. This study evaluated the foliar content of nutrients (N, P, K, and Ca) and proline, antioxidant activity, growth, water use efficiency, and yield of cowpea cultivars subjected to two irrigation depths (50% and 100% of crop evapotranspiration) and a foliar application of silicon (orthosilicic acid). A field experiment was conducted in a split-plot scheme using the randomized block design with four replications in a semi-arid region of northeastern Brazil. Silicon supplementation increased the foliar contents of N, P, and Ca; stimulated proline synthesis; and enhanced the activity of the SOD, CAT, and APX enzymes. These changes promoted growth, improved water use efficiency, and increased crop yield. The results indicate that foliar silicon application mitigates the effects of water deficit in cowpea plants while enhancing crop performance under full irrigation (100% of crop evapotranspiration), leading to higher yields even under favorable water conditions.

## 1. Introduction

Cowpea (*Vigna unguiculata* L. Walp.) is an annual legume native to Africa and cultivated worldwide, with a particular emphasis on tropical regions [1]. This crop is recognized for its high nutritional potential and plays a fundamental role in food security across various countries, serving as a rich source of proteins, carbohydrates, vitamins, and minerals for human consumption and an important forage source for animal feed [2]. Another relevant aspect is its contribution to soil ecological balance as it promotes biological nitrogen fixation associated with nodulating bacteria [3].

In Brazil, cowpea is widely cultivated in the northeastern region, playing a significant role in job creation and income generation for family farming [4]. In the 2023/2024 growing season, national cowpea production reached 691.8 thousand tons, accounting for approximately 21.3% of the country’s total bean production [5]. However, despite its socio-economic importance, cowpea still exhibits a relatively low yield compared to other bean species, primarily due to the predominance of cultivation under limited water availability [5,6,7].

Water deficit is one of the main abiotic factors negatively impacting global agriculture, particularly in semi-arid regions where cowpea cultivation is predominant [3]. This type of stress affects various morphological, physiological, biochemical, and molecular traits, leading to significant reductions in crop growth and yield [6,7,8,9]. Management strategies to increase plants’ tolerance to water deficit have been widely studied in this context.

Among the promising alternatives, silicon (Si) application stands out as a beneficial element that, although not essential, has great potential to mitigate abiotic stresses such as drought. Studies indicate that silicon can stimulate the accumulation of osmoregulators, enhance antioxidant capacity, optimize photosynthesis, improve nutrient uptake, and preserve the structural integrity of plant cells, resulting in benefits for various crops [10,11,12,13,14,15,16]. However, the mechanisms involved in these processes are not yet fully elucidated. Despite these advancements, knowledge gaps remain regarding the impact of silicon fertilization on cowpea genotypes under water deficit conditions in semi-arid environments.

Understanding the mechanisms of water deficit tolerance in this species is essential, particularly regarding foliar silicon application combined with different irrigation depths. Spraying silicon on leaves aims to compensate for low root uptake, which can occur in soils with naturally limited silicon availability, increasing its absorption and enhancing silicon’s beneficial effects [17]. By activating physiological, biochemical, and molecular processes, this practice can increase plants’ tolerance to water stress, providing valuable insights for optimizing management practices. As a result, improving water use efficiency and, consequently, enhancing crop yield and farmers’ profitability are possible [16].

Given this context, this study aimed to evaluate the effects of foliar silicon application on cowpea cultivars subjected to both deficit and adequate irrigation regimes. The focus was on analyzing foliar nutrient concentrations, proline accumulation, antioxidant enzyme activity, vegetative growth, water use efficiency, and crop yield performance.

## 2. Results

### 2.1. Nutrient Accumulation, Proline Content, and Antioxidant Activity

Table 1 summarizes the analysis of variance for variables related to nutrient accumulation, free proline content, and antioxidant enzyme activity in cowpea genotypes under different irrigation depths and foliar silicon applications. The three-way interaction among irrigation depths, silicon, and cultivars significantly only influenced the shoot dry mass and the activity of the enzyme ascorbate peroxidase (APX). The interaction between irrigation depths and cultivars affected the foliar nitrogen and phosphorus concentrations and the proline content. Silicon, when considered independently, had a significant effect on the foliar nitrogen, phosphorus, and calcium contents, in addition to the proline content and the activities of superoxide dismutase (SOD) and catalase (CAT) enzymes. On the other hand, except for proline, the irrigation depths significantly impacted all analyzed variables, highlighting the influence of water availability on the crop.

The foliar application of orthosilicic acid resulted in significant benefits in the accumulation of nitrogen, phosphorus, and calcium in cowpea leaf tissues, highlighting the role of silicon in improving plant mineral nutrition (Figure 1a–c). The leaf nitrogen content increased by 21% in plants that received silicon application (Figure 1a), while the phosphorus content showed a 38% increase in treated plants compared to untreated ones (Figure 1b). Similarly, the leaf calcium content increased by 17% in plants supplemented with silicon (Figure 1c).

Among the cultivars, the nitrogen content ranged depending on irrigation depths. Under an irrigation depth of 50% of ETc, the cultivars BRS Pujante and BRS Pajeú showed an average increase of 23% in foliar N accumulation compared to BRS Novaera. However, under an irrigation depth of 100% of ETc, only the cultivar BRS Pajeú stood out, with a foliar nitrogen content 29% higher than the average of the other cultivars (Figure 1d). Regarding phosphorus accumulation, differences were evident only under full irrigation conditions, with BRS Pujante exhibiting the highest foliar phosphorus content at 100% of ETc, surpassing the average of the other cultivars by 45% (Figure 1e).

Regarding irrigation depths, all variables associated with nutrient accumulation showed higher mean values in plants irrigated at 100% of ETc than those subjected to water deficit (Figure 1d–g). The foliar N (Figure 1d) and P (Figure 1e) contents were, on average, 41% and 53% higher, respectively, while the Ca content increased by 39% (Figure 1f) in adequately irrigated plants compared to those receiving a lower irrigation depth. Although silicon application did not significantly influence the potassium content (Table 1), full irrigation resulted in a 35% increase in the foliar K content compared to plants under water deficit conditions (Figure 1g).

The irrigation depths did not affect the free proline content in cowpea leaves but ranged among cultivars and in response to foliar silicon application (Figure 2). Silicon supplementation significantly increased proline concentrations in all evaluated cultivars, with 100%, 46%, and 166% increases for BRS Novaera, BRS Pujante, and BRS Pajeú, respectively, compared to untreated plants. Additionally, regardless of silicon application, the cultivar BRS Pujante exhibited the highest foliar proline levels.

The data on antioxidant enzyme activity in cowpea cultivars subjected to different irrigation depths and leaf silicon applications are presented in Figure 3. SOD activity increased under water deficit, regardless of the cultivar, with a 33% increase in plants irrigated at 50% of ETc compared to those receiving adequate irrigation (Figure 3a). Foliar silicon application also enhanced antioxidant enzyme activity, with a 6% increase in SOD activity (Figure 3b) and a 17% increase in CAT activity (Figure 3c) compared to untreated plants. Among the evaluated cultivars, BRS Pujante exhibited the highest CAT activity (Figure 3d).

APX activity was influenced by the three-way interaction, with the effects of silicon application varying among cultivars and water conditions (Figure 3e). Under water deficit (50% of ETc), the BRS Novaera cultivar exhibited a 62% increase in APX activity in response to silicon application compared to untreated plants, while no significant differences were observed in the other cultivars. Under full irrigation conditions, silicon also had beneficial effects, with variations among cultivars. When irrigated at 100% of ETc, BRS Pujante and BRS Pajeú showed increases of 137% and 354%, respectively, with silicon application, whereas BRS Novaera was unaffected.

### 2.2. Growth and Production Components

The summary of the analysis of variance for variables related to growth, water use efficiency (WUE), and yield components of cowpea genotypes under different irrigation depths and silicon application is presented in Table 2. The three-way interaction among irrigation depths, silicon application, and cultivars only significantly influenced the shoot dry mass (SDM). The main branch length (MBL) was independently affected by both foliar silicon application and irrigation depths, demonstrating the influence of these factors on vegetative growth. Water use efficiency (WUE) was significantly influenced by the irrigation × silicon and silicon × cultivar interactions, suggesting that plant responses to silicon depend on both water availability and the genetic characteristics of the cultivars. The number of pods per plant (NPP) ranged due to irrigation depths, silicon, and cultivars. Meanwhile, grain yield (Yield) was affected by irrigation depths and the silicon × cultivar interaction, indicating that genotypic differences may condition the productive response of plants to silicon.

Foliar silicon supplementation resulted in significant gains in cowpea vegetative growth (Figure 4). The main branch length increased by an average of 55% in silicon-treated plants compared to untreated ones (Figure 4a). Additionally, the MBL was significantly higher under full irrigation, with a 55% increase compared to plants subjected to water deficit (Figure 4b). Water restriction (50% of ETc) led to reductions in shoot dry mass across all evaluated cultivars; however, silicon application mitigated these losses and promoted increases in SDM under both water deficit and full irrigation conditions (Figure 4c). Under 50% of ETc, SDM increased by 214%, 73%, and 45% for the cultivars BRS Novaera, BRS Pujante, and BRS Pajeú, respectively. Under 100% of ETc, the increases reached 86%, 48%, and 344% for the same cultivars, highlighting the influence of silicon under different water conditions, albeit with varying intensities among the evaluated cultivars.

The results related to the water use efficiency (WUE) of cowpea genotypes subjected to different irrigation depths and foliar applications of orthosilicic acid are presented in Figure 5. Regardless of the genotype or water conditions, foliar silicon application increased water use efficiency. Under 50% of ETc, silicon supplementation increased the WUE by 41% compared to untreated plants (Figure 5a), demonstrating its potential in mitigating the effects of water deficit. Under adequate irrigation, silicon application led to a 32% increase in WUE. The BRS Novaera cultivar also exhibited the highest WUE across all evaluated conditions (Figure 5b).

Mitigating effects of water deficit through silicon application were also observed in the number of pods per plant (NPP) of the BRS Novaera cultivar (Figure 6). When irrigated at 50% of ETc, this cultivar showed a 23% increase in NPP with silicon application compared to untreated plants. Silicon supplementation also enhanced pod production under full irrigation, resulting in a 30% increase in NPP when BRS Novaera received 100% ETc (Figure 6). On the other hand, in the absence of silicon application, although BRS Pujante and BRS Pajeú exhibited lower NPP values, they did not experience significant reductions in response to water deficit, suggesting a higher degree of tolerance to the imposed conditions.

Water deficit significantly reduced the dry grain yield of cowpeas regardless of the cultivar (Figure 7a). Plants subjected to 50% of ETc showed an average yield reduction of 22% compared to those receiving full irrigation at 100% of ETc, highlighting the negative impact of water restrictions on crop yield. Despite this limitation, foliar silicon application led to substantial yield gains in all cultivars regardless of water availability. The increases were 25% in BRS Novaera, 83% in BRS Pujante, and 18% in BRS Pajeú compared to plants that did not receive silicon (Figure 7b), demonstrating the effectiveness of silicon in optimizing crop yield. Among the evaluated cultivars, BRS Novaera exhibited the highest yield values with and without foliar silicon supplementation (Figure 7b). When treated with silicon, this cultivar reached a grain yield of 2952 kg ha^−1^, standing out as the most responsive to silicon application and reinforcing its high yield potential.

## 3. Discussion

According to the obtained results, the cowpea cultivars that received foliar applications of orthosilicic acid exhibited higher foliar nitrogen (Figure 1a), phosphorus (Figure 1b), and calcium (Figure 1c) contents, supporting studies that highlight the role of silicon in nutrient uptake and accumulation in plants [17,18,19,20,21,22,23]. Among these nutrients, the increase in foliar N content may be associated with silicon’s influence on nitrogen metabolism regulation in plants [18,22]. Hao et al. [22] reported that nano-SiO_2_ application in maize stimulated the activity of enzymes involved in nitrogen metabolism, such as glutamine synthetase, glutamate decarboxylase, and glutamate dehydrogenase, promoting greater efficiency in nitrogen assimilation in both leaves and roots.

In addition to silicon’s influence on nitrogen enzyme metabolism, Mali and Aeri [18] observed that soil silicon application, up to certain doses, stimulated root nodulation in cowpeas, promoting greater biological nitrogen fixation by symbiotic bacteria. This effect may be related to strengthening plant–microorganism interactions, possibly due to the increased availability of organic compounds exuded by roots in response to silicon [23]. Thus, we hypothesize that silicon’s role in modulating nitrogen uptake and assimilation and its positive impact on nodulation and biological fixation may have been a key factor in increasing the foliar nitrogen content in cowpeas [18]. This effect can be particularly relevant in soils with low nitrogen availability, where biological fixation is essential for adequate nutrient supply.

In addition to nitrogen, phosphorus also showed a significant increase in silicon-treated plants (Figure 1b), suggesting a positive effect of Si on P acquisition and translocation, possibly through the regulation of genes involved in nutrient transport, as observed by Kostic et al. [23] in wheat. In their study, soil silicon application increased the expression of the TaPHT1;1 and TaPHT1;2 genes, which encode inorganic phosphate transporters, promoting phosphorus uptake by plants. Furthermore, these authors reported that silicon enhanced the exudation of carboxylates by roots, potentially increasing phosphorus availability in the rhizosphere and contributing to greater nutrient efficiency. It is hypothesized that a similar mechanism may have contributed to the higher P levels observed in silicon-treated cowpea plants.

The increase in foliar calcium contents (Figure 1c) further reinforces the role of silicon in nutrient acquisition by cowpeas. This effect may be associated with silicon’s ability to mitigate early-stage plant stress by promoting cell membrane stability and stimulating the activity of H^+^-ATPase, an essential enzyme for ion transport [18,19]. The regulation of this mechanism may explain the higher Ca accumulation in cowpea leaves observed in the present study.

Although the benefits of silicon in nitrogen, phosphorus, and calcium uptake are evident, the potassium contents in the treated plants were not significantly influenced by Si application (Table 1), representing a noteworthy finding. Potassium is essential for transpiration control, stomatal guard cell turgor regulation, carbohydrate translocation, and various enzymatic reactions [24]. While some studies suggest that silicon may enhance potassium uptake in certain crops, its effects are still not fully understood [19]. Therefore, other factors may have contributed to the observed results. Narwal et al. [25] reported antagonism between K and Ca in cowpea plants, where a high concentration of one ion can reduce the uptake of the other. In sorghum, Chen et al. [26] found that silicon application did not influence K levels in the shoots or roots but increased the photosynthetic rate and plant biomass. A molecular analysis conducted by the same authors in Arabidopsis [26] showed that Si application did not induce the expression of HAK5 and AKT1 genes responsible for K^+^ uptake.

In the literature, there is extensive evidence that the effects of silicon on nutrient uptake in plants vary widely depending on factors such as plant species, environmental conditions, and the mode of application of the element, among others [18,19,21,27]. Mali and Aery [18] observed that soil Si addition increased foliar P and Ca levels up to a certain threshold in cowpeas, but excessive doses reduced nutrient accumulation in plants. In contrast, Jam et al. [27] reported that in Carthamus tinctorius, the foliar application of Si at 2.5 mM increased the Ca levels only in plants grown under irrigation with high-quality water, with no significant effect being found under salt stress. These findings suggest that specific environmental factors in the cultivation system may condition the impact of Si.

Proline is a key osmolyte in response to abiotic stresses, playing a role in osmoregulation, the protection against oxidative damage, the regulation of stress tolerance-related gene expression, and the maintenance of water balance, among other functions [28,29,30,31]. In the present study, a significant increase in proline levels was observed in silicon-treated plants (Figure 2), suggesting that this element may have played a crucial role in activating osmoprotection mechanisms. The rise in proline levels in response to silicon may be associated with its ability to mitigate the effects of water stress by reducing cellular water loss and protecting macromolecules from denaturation and peroxidation [31,32].

Although irrigation depths did not have a significant effect on the proline content (Table 1), the increase promoted by silicon is relevant for enhancing plant tolerance to abiotic stresses, especially considering that the experiment was conducted in a semi-arid region characterized by high temperatures, intense solar radiation, and low relative humidity (Figure 8). Santos et al. [31] also reported increased proline levels in cowpea cv. BRS Guariba in response to foliar silicon application, conferring greater tolerance to water stress. However, the specific mechanisms by which silicon stimulates proline synthesis are not fully understood, highlighting the need for further studies.

Based on the results obtained for antioxidant activity (Figure 3), it is evident that the response of cowpea cultivars is modulated by both water conditions and silicon application, which aligns with previous studies [13,31,32]. The increase in SOD activity under water deficit (Figure 3a) confirms that water deficiency intensifies the production of reactive oxygen species, thereby stimulating plant antioxidant defense mechanisms [31]. However, silicon application also influenced the antioxidant response in cowpea, increasing SOD (Figure 3b) and CAT (Figure 3c) activities, demonstrating the effectiveness of this element in mitigating oxidative damage. SOD plays a key role in converting superoxide radicals (O_2_^−^) into hydrogen peroxide (H_2_O_2_), while CAT catalyzes the decomposition of H_2_O_2_ into water and oxygen, preventing the accumulation of reactive oxygen species and subsequent lipid peroxidation in cell membranes [33].

Notably, when treated with silicon, the BRS Novaera cultivar exhibited a substantial 62% increase in APX activity under water deficit (Figure 3e). APX plays a crucial role in H_2_O_2_ elimination through the ascorbate–glutathione cycle, a fundamental mechanism for oxidative homeostasis under stress conditions [33]. These results reinforce that silicon is key in regulating oxidative stress in cowpeas cultivated under adverse conditions, as observed in other studies [13,30,31,32]. Similarly, Santos et al. [31] and Silva et al. [32] also reported increased antioxidant enzyme activities in cowpeas under water stress, confirming the importance of Si in regulating these defense mechanisms.

The nutritional, osmoregulatory, and antioxidant improvements promoted by silicon significantly enhanced cowpea’s vegetative growth, resulting in substantial increases in the main branch length (Figure 4a) and shoot biomass (Figure 4c). Additionally, studies indicate that the benefits of Si may also be associated with increased photosynthetic activity, silica deposition in the leaf epidermis, the strengthening of the plasma membrane, and a reduced transpiration rate, all of which collectively enhance plant development [13,31,32,34,35]. The notable increases in shoot dry mass, particularly under water deficit conditions (Figure 4c), highlight silicon’s ability to mitigate the negative effects of water restriction in cowpeas. These findings align with those of Silva et al. [32], who reported that foliar silicon application in potassium silicate alleviates water stress and improves growth characteristics in cowpea cultivars.

In addition to biochemical benefits and plant growth improvements, silicon application significantly enhanced the evaluated yield parameters, as demonstrated in Figure 5, Figure 6 and Figure 7. The increase in WUE in plants subjected to 50% of ETc and treated with silicon (Figure 5a) highlights the role of this element in maintaining yield even under water restrictions. Additionally, the increase in WUE under full irrigation with silicon application (Figure 5a) demonstrates that the benefits of this element are not limited to stress conditions but also contribute to more efficient water use under adequate moisture availability. The superior performance of the BRS Novaera cultivar across all evaluated conditions (Figure 5b) reinforces the influence of genetic factors on water use efficiency, possibly associated with intrinsic traits that promote higher yield with lower water consumption.

Silicon application also minimized the effects of water deficit on pod production, with the BRS Novaera cultivar standing out for its superior performance under all evaluated conditions (Figure 6). The increase in NPP promoted by silicon application under irrigation at 50% of ETc demonstrates this element’s ability to mitigate the impacts of water deficiency, contributing to the maintenance of reproductive structure development even under adverse conditions. Furthermore, the increase in NPP under full irrigation indicates that silicon reduces the effects of water stress and maximizes yield potential when adequate water availability is ensured.

Regardless of the water regime, silicon application significantly increased yield in the evaluated cultivars (Figure 7a). The BRS Novaera cultivar, which reached a yield of 2952 kg ha^−1^ with silicon application (Figure 7b), stood out for its high yield potential and strong responsiveness to silicon, establishing itself as a promising option for cultivation in semi-arid regions. Furthermore, the observed yield values far exceeded the national average for the crop, estimated at 542 kg ha^−1^ in the 2023/2024 growing season [5], reinforcing the feasibility of using silicon as a strategy to optimize cowpea agronomic performance, particularly in systems subject to water restrictions.

The efficiency of the foliar application of orthosilicic acid has been widely debated. According to Laane [16], foliar application can also be an effective strategy, as large amounts of silicates must be applied to the soil to release monosilicic acid, the only form assimilable by plants. This process depends on factors such as pH, soil composition, and microbial activity, which can limit the immediate availability of silicon for root uptake. In contrast, Si supplied via spraying is already bioavailable, allowing for faster and more direct absorption by foliar tissues. However, according to the authors, the specific mechanism by which monosilicic acid penetrates the leaf epidermis and is redistributed within plant tissues is not yet fully elucidated. Further studies are needed to understand the factors regulating this absorption and subsequent translocation, particularly in different species and environmental conditions. Nevertheless, we consider foliar silicon application to be an effective method for supplying silicon to cowpea.

## 4. Materials and Methods

### 4.1. Location, Treatments, and Statistical Design

A field experiment was conducted between 17 September and 18 November 2021 at the Center for Human and Agricultural Sciences of the State University of Paraíba, located in the municipality of Catolé do Rocha, PB, Brazil (6°20′ 38″ S, 37°44′ 48″ W, and altitude of 275 m). The climate of the region is classified as BSh-type, a hot semi-arid climate with summer rainfall [36]. The climatic data recorded during the experiment are presented in Figure 8.

The experiment was arranged in a split-plot design, with the plots corresponding to two irrigation depths (50% and 100% of crop evapotranspiration—ETc) and the subplots consisting of the combination of three cowpea cultivars (BRS Novaera, BRS Pujante, and BRS Pajeú) and the application or not of foliar silicon fertilization (with application and control without application). All the cultivars evaluated are well adapted to a wide range of climates and are recommended for cultivation in Northeast Brazil, particularly in semi-arid environments. BRS Novaera exhibits a semi-erect growth habit, while BRS Pujante and BRS Pajeú have a semi-prostrate growth habit. A randomized block design was adopted with four replications. The main plot consisted of six subplots spaced 0.5 m apart. Each subplot contained three planting rows, with 10 plants per row (spacing of 0.1 m between plants × 0.5 m between rows), totaling 30 plants per subplot.

Silicon solutions were prepared by dissolving orthosilicic acid [Si(OH)_4_] (pH 7.0) in water, resulting in a 10 mg Si L^−1^ concentration, which was adapted from Silva et al. [32]. Three foliar applications were performed on the abaxial and adaxial leaf surfaces 19, 35, and 42 days after sowing (DAS), as shown in Figure 8. In each application, spraying was conducted using approximately 35, 70, and 90 mL m^−2^ of the solution, respectively.

### 4.2. Irrigation Management

The plants were irrigated daily with water from a well, which had the following characteristics: electrical conductivity = 1.01 dS m^−1^; pH = 6.9; concentrations of K^+^, Ca^2+^, Mg^2+^, Na^+^, Cl^−^, HCO_3_^−^, and SO_4_^2−^ of 1.21, 2.5, 1.48, 6.45, 8.1, 2.75, and 0.18 mmol_c_ L^−1^, respectively; and sodium adsorption ratio = 4.57 (mmol L^−1^)^1^/^2^. Irrigation was performed via a drip system using drip tapes with emitters spaced every 0.2 m and a flow rate of 1.6 L h^−1^, operating at a service pressure of 0.1 MPa.

Crop evapotranspiration (ETc) was determined by multiplying reference evapotranspiration (ETo, mm d^−1^) by crop coefficients (Kc), which varied according to the plant phenological stages. ETo was estimated based on Class ‘A’ pan evaporation, corrected using a coefficient of 0.75. The plants’ consumptive water use (CU) was considered, assuming 100% of the wetted area (P). To determine the daily net irrigation depth (DLD), the following equation was used: DLD = CU × P/100 (mm d^−1^). The applied irrigation depths corresponded to 50% and 100% of the DLD, with adjustments based on irrigation time. The irrigation depths applied at each phenological stage of the plants are presented in Table 3, allowing for an assessment of water management over the crop development cycle.

### 4.3. Soil Characteristics and Preparation of Experimental Area

The soil in the experimental area was classified as an Entisol (Fluvent) [37], with a sandy clay loam texture, and had the following physical properties: 831.5, 100.0, and 68.5 g kg^−1^ of sand, silt, and clay, respectively; soil bulk density = 1.53 g cm^−3^; particle density = 2.61 g cm^−3^; total porosity = 0.42 m^3^ m^−3^; flocculation degree = 1000 kg dm^−3^; and moisture contents of 65, 49, and 28 g kg^−1^ at tensions of −0.01, −0.03, and −1.50 MPa, respectively. Regarding fertility, the soil had the following characteristics: pH = 6.0; P = 16.63 mg dm^−3^; concentrations of K^+^, Ca^2+^, Mg^2+^, and Na^+^ of 0.08, 1.09, 1.12, and 0.05 cmol_c_ dm^−3^, respectively; sum of exchangeable bases = 2.34 cmol_c_ dm^−3^; H^+^ + Al^3+^ = 1.24 cmol_c_ dm^−3^; Al^3+^ = 0 cmol_c_ dm^−3^; cation exchange capacity (CEC) = 3.58 cmol_c_ dm^−3^; base saturation (V) = 65.36%; and organic matter content = 13.58 g kg^−1^.

Soil preparation consisted of plowing to a depth of 50 cm. The fertilization at the sowing furrow and in top-dressing followed the recommendations of Cavalcanti [38]. Before sowing, the soil was irrigated until it reached field capacity moisture. Subsequently, sowing was carried out using two seeds per hole at a depth of 3 cm. Five days after seedling emergence, thinning was performed, leaving one plant per hole.

### 4.4. Experimental Analysis

For foliar nutrient analysis, four leaves from the middle section of the three central plants in each subplot were collected 45 days after sowing (DAS). After being washed with distilled water, the leaves were dried in a forced-air circulation oven at 60 °C until reaching a constant mass, ground in a stainless steel knife mill (Willey-type), and stored in hermetically sealed containers for the determination of nitrogen (N), phosphorus (P), potassium (K), and calcium (Ca) concentrations, following the methodologies proposed by Silva [39].

Nitrogen (N) content was determined using the Kjeldahl method (dry digestion); phosphorus content was measured by molybdenum blue spectrophotometry; potassium (K) content was determined by flame photometry; and calcium (Ca) content was analyzed using atomic absorption spectrometry at a wavelength of 422.7 nm.

At 45 DAS, the main stem length (MSL) was also analyzed, measured from the base of the stem to the base of the youngest leaf. During the same period, the shoot of three plants per subplot was collected, placed in paper bags, and dried in a forced-air circulation oven at 65 °C. After reaching constant mass, the material was weighed using a precision balance (0.0001 g), and shoot dry mass (SDM) was determined.

At 45 DAS, the youngest fully expanded leaves from three plants per subplot were also collected for the analysis of proline content and the activities of antioxidant enzymes: superoxide dismutase (SOD), catalase (CAT), and ascorbate peroxidase (APX). The plant material was collected in the morning, stored in ice-filled containers, and immediately transported to the laboratory, where it was kept in a freezer until analysis.

Proline content (μmol g^−1^ FM—fresh matter) was determined using the colorimetric method described by Bates et al. [40] and modified by Bezerra Neto and Barreto [41]. Fresh leaf tissue (250 mg) was macerated in 5 mL of 3% sulfosalicylic acid and centrifuged at 2000 rpm for 10 min. The supernatant was collected into 2.5 mL tubes and stored under refrigeration. Proline quantification was performed by spectrophotometry, with absorbance taken at 520 nm.

For the determination of antioxidant enzyme activity, the same enzymatic extract was used, prepared from 200 mg of fresh leaf tissue, homogenized using mortar and pestle with 3.0 mL of potassium phosphate buffer (100 mM, pH 7.0) containing 1 mM EDTA. SOD activity was determined based on the ability of the enzymatic extract to inhibit the photoreduction of nitro blue tetrazolium (NBT) according to the methodology described by Beauchamp and Fridovich [42]. Absorbance readings were taken at 560 nm, and the results were expressed as AU g^−1^ FM.

CAT activity was determined according to the method described by Kar and Mishra [43] based on the decrease in absorbance at 240 nm The results were expressed as µmol H_2_O_2_ min^−1^ g^−1^ FM. For de assay, 100 µL of the enzymatic extract was added to 2.9 mL of the reaction mixture, which consisted of 500 µL of H_2_O_2_ (59 mM), 1.5 mL of potassium phosphate buffer (0.05 M, pH 7.0), and 400 µL of deionized water, maintained at 30 °C.

APX activity was determined following the method proposed by Nakano and Asada [44], based on ascorbic acid consumption, measured by the decrease in absorbance at 290 nm using a quartz cuvette. The reaction mixture consisted of 50 mM potassium phosphate buffer (pH 6.0) and 0.8 mM ascorbic acid. The reaction was initiated by adding 200 µL of 2 mM H_2_O_2_ to 300 µL of the enzymatic extract. Results were expressed as mmol ASC (ascorbate) min^−1^ g^−1^ FM.

At the end of the cycle, between 56 and 62 DAS, pod harvesting was carried out on the remaining plants in the subplot, and we recorded the number of pods per plant (NPP) and grain yield (kg ha^−1^). Additionally, water use efficiency (WUE) was evaluated based on the ratio between grain yield (kg ha^−1^) and the irrigation depth (mm), as described by Souza et al. [45].

### 4.5. Statistical Analysis

The obtained data were subjected to normality and homoscedasticity tests using the Shapiro–Wilk method (*p* < 0.05) and Bartlett test (*p* < 0.05), respectively. Subsequently, they were analyzed using the analysis of variance (ANOVA) with the F-test (*p* < 0.05). Mean comparisons were performed using the Tukey test (*p* < 0.05).

## 5. Conclusions

The results of this study highlight foliar silicon supplementation as an effective strategy for mitigating the impacts of water deficit in cowpea, promoting improvements in nutritional, osmoprotective, and antioxidant properties. These effects enhanced plant growth, water use efficiency, and crop yield, reinforcing the role of silicon in plant adaptation to water stress conditions. Furthermore, the benefits observed even under full irrigation indicate that silicon mitigates the effects of water limitation and contributes to maximizing the crop’s yield potential under adequate water supply conditions.

## Figures and Tables

**Figure 1 plants-14-01241-f001:**
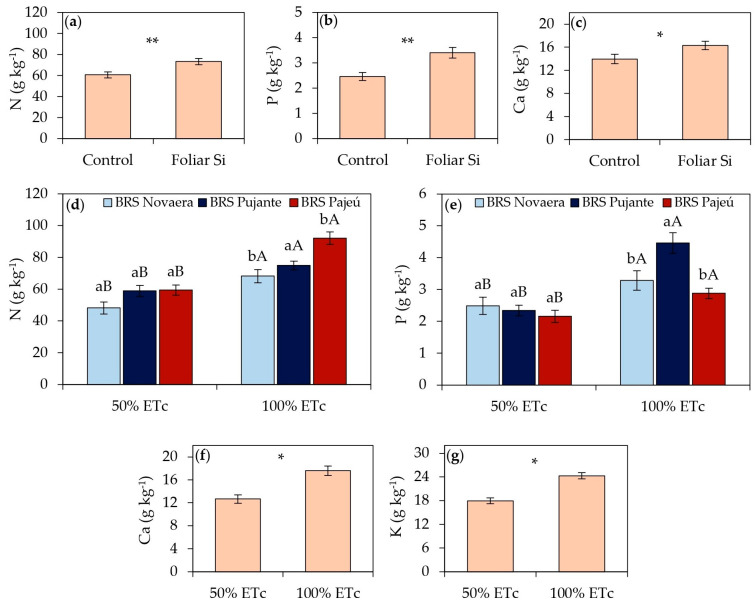
Foliar contents of N (**a**,**d**), P (**b**,**e**), Ca (**c**,**f**), and K (**g**) in cowpea cultivars subjected to different irrigation depths and foliar silicon applications. ** and * indicate significance at *p* < 0.01 and *p* < 0.05, respectively, according to F-test. Lowercase letters indicate differences among cultivars, and uppercase letters indicate differences among irrigation depths, according to Tukey’s test. Bars represent standard error of mean.

**Figure 2 plants-14-01241-f002:**
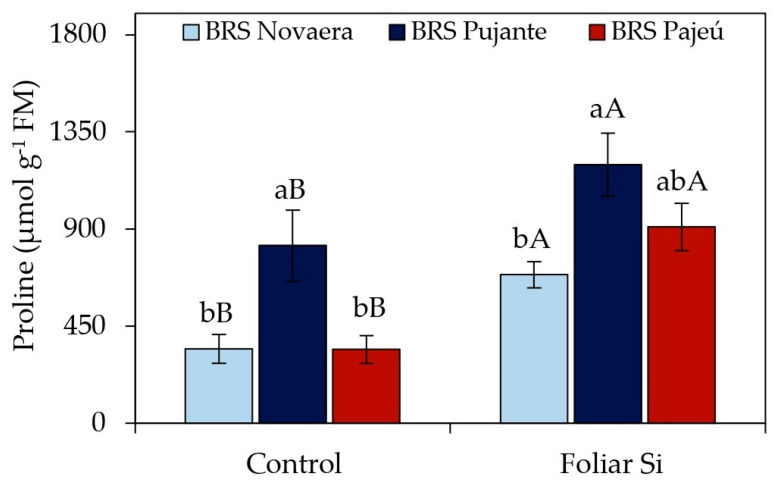
Effect of silicon × cowpea cultivar interaction on free proline content in leaves. Lowercase letters indicate differences among cultivars, and uppercase letters indicate differences between presence and absence of foliar silicon application, according to Tukey’s test. Bars represent standard error of mean.

**Figure 3 plants-14-01241-f003:**
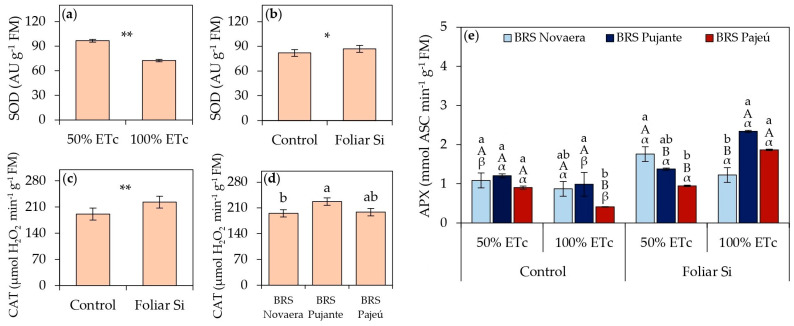
Activities of enzymes superoxide dismutase (SOD) (**a**,**b**), catalase (CAT) (**c**,**d**), and ascorbate peroxidase (APX) (**e**) in cowpea cultivars subjected to different irrigation depths and foliar silicon applications. ** and * indicate significance at *p* < 0.01 and *p* < 0.05, respectively, according to F-test. Lowercase letters indicate differences among cultivars; uppercase letters indicate differences among irrigation depths; and Greek letters indicate differences between presence and absence of foliar silicon application, according to Tukey’s test. Bars represent standard error of mean.

**Figure 4 plants-14-01241-f004:**
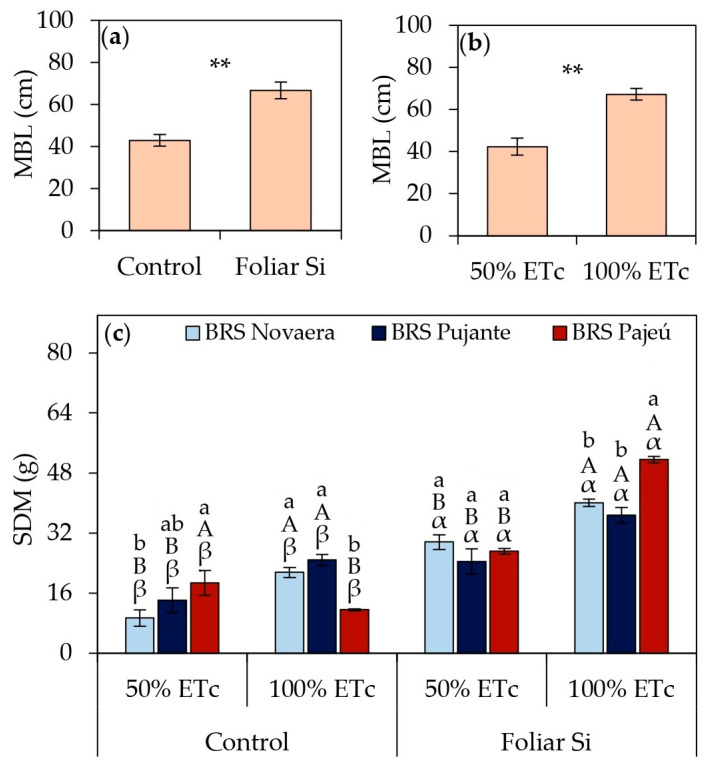
Main branch length (MBL) (**a**,**b**) and shoot dry mass (SDM) (**c**) of cowpea cultivars subjected to different irrigation depths and foliar silicon application. ** indicates significance at *p* < 0.01 according to F-test. Lowercase letters indicate differences among cultivars; uppercase letters indicate differences among irrigation depths; and Greek letters indicate differences between presence and absence of foliar silicon application according to Tukey’s test. Bars represent standard error of mean.

**Figure 5 plants-14-01241-f005:**
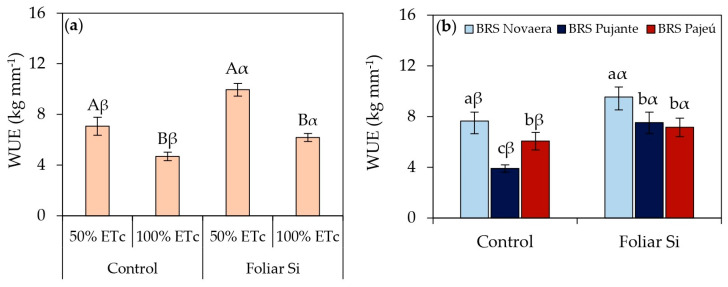
(**a**) Effect of irrigation depths × silicon interaction and (**b**) effect of silicon × cultivar interaction on yield. Lowercase letters indicate differences among cultivars; uppercase letters indicate differences among irrigation depths; and Greek letters indicate differences between presence and absence of foliar silicon application according to Tukey’s test. Bars represent standard error of mean.

**Figure 6 plants-14-01241-f006:**
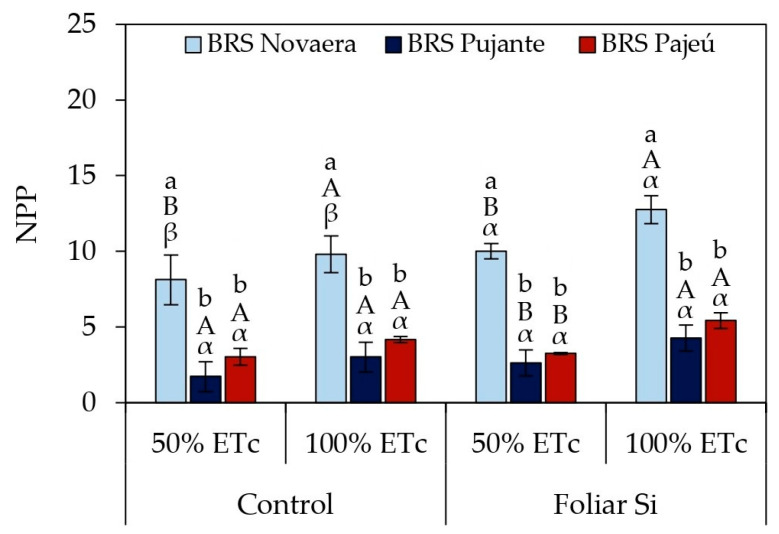
Effect of irrigation × silicon × cultivar interaction on number of pods per plant (NPP) in cowpea. Lowercase letters indicate differences among cultivars; uppercase letters indicate differences among irrigation depths; and Greek letters indicate differences between presence and absence of foliar silicon application according to Tukey’s test. Bars represent standard error of mean.

**Figure 7 plants-14-01241-f007:**
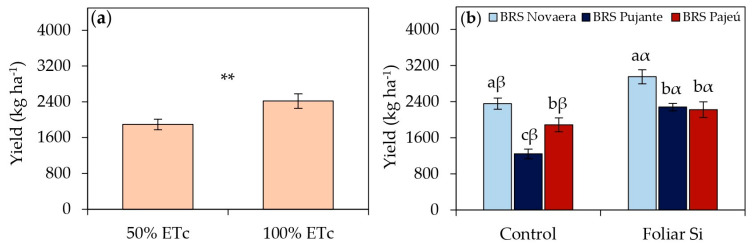
(**a**) Grain yield of cowpea cultivars subjected to different irrigation depths and (**b**) effect of silicon × cultivar interaction on yield. ** indicates significance at *p* < 0.01 according to F-test. Lowercase letters indicate differences among cultivars, and Greek letters indicate differences between presence and absence of foliar silicon application according to Tukey’s test. Bars represent standard error of mean.

**Figure 8 plants-14-01241-f008:**
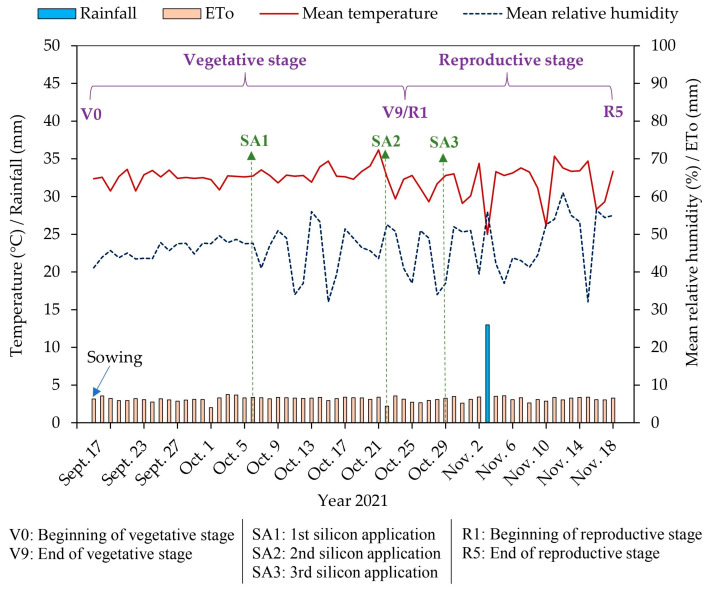
A graphical representation of the weather conditions recorded in the experimental area during the experiment and the time representation of the main experimental events.

**Table 1 plants-14-01241-t001:** A summary of the analysis of variance for the leaf contents of nitrogen (N), phosphorus (P), potassium (K), and calcium (Ca); the free proline content (Proline); and the activity of the enzymes superoxide dismutase (SOD), catalase (CAT), and ascorbate peroxidase (APX) in cowpea cultivars under different irrigation depths and foliar silicon applications.

Source of Variation	DF	Mean Square							
		N	P	K	Ca	Proline	SOD	CAT	APX
Block	3	24.32 ^ns^	0.078 ^ns^	47.91 ^ns^	23.47 ^ns^	89,960.80 ^ns^	6.24 ^ns^	2198.51 ^ns^	0.09 ^ns^
Irrigation depth (ID)	1	6298.41 **	17.62 **	484.75 *	291.60 *	845,365.35 ^ns^	7020.90 **	5257.19 ^ns^	0.05 ^ns^
Error A	3	34.46	0.11	47.55	15.63	88,566.39	24.47	701.05	0.09
Silicon (Si)	1	1913.19 **	10.59 **	24.39 ^ns^	65.26 *	2,205,125.63 **	290.96 *	12,233.66 **	5.46 **
Cultivar (Cul)	2	1241.71 **	3.14 **	15.28 ^ns^	22.004 ^ns^	1,078,779.33 **	40.04 ^ns^	4913.51 *	0.79 **
ID × Si	1	0.02 ^ns^	0.34 ^ns^	78.23 ^ns^	23.78 ^ns^	316,418.53 ^ns^	0.45 ^ns^	1029.15 ^ns^	1.72 **
ID × Cul	2	304.26 **	2.47 **	64.44 ^ns^	5.91 ^ns^	1,043,897.35 **	79.93 ^ns^	1503.52 ^ns^	0.62 **
Si × Cul	2	33.94 ^ns^	0.60 ^ns^	6.08 ^ns^	1.75 ^ns^	58,459.35 ^ns^	7.11 ^ns^	1176.37 ^ns^	0.08 ^ns^
ID × Si × Cul	2	29.72 ^ns^	0.25 ^ns^	14.96 ^ns^	1.51 ^ns^	247,645.19 ^ns^	55.12 ^ns^	622.71 ^ns^	0.89 **
Error B	30	1249.33	0.23	47.31	12.91	76,424.02	60.15	1193.64	0.07
CV (A) (%)	-	8.76	11.49	32.61	26.13	41.47	5.85	12.79	24.71
CV (B) (%)	-	9.63	16.57	32.53	23.75	38.52	9.17	16.68	22.49

^ns^—not significant; **—significant at *p* ≤ 0.01; *—significant at *p* ≤ 0.05 according to the F-test; DF—degrees of freedom; CV—coefficient of variation.

**Table 2 plants-14-01241-t002:** Summary of variance analysis for main branch length (MBL), shoot dry mass (SDM), water use efficiency (WUE), number of pods per plant (NPP), and grain yield (Yield) in cowpea cultivars under different irrigation depths and foliar silicon applications.

Source of Variation	DF	Mean Square				
		MBL	SDM	WUE	NPP	Yield
Block	3	43.05 ^ns^	61.51 ^ns^	0.98 ^ns^	1.31 ^ns^	42,760.63 ^ns^
Irrigation depth (ID)	1	7413.99 **	1318.69 **	113.65 **	38.10 **	3,298,685.88 *
Error A	3	75.50	4.75	2.47	0.88	190,185.27
Silicon (Si)	1	6746.20 **	3984.53 **	57.64 **	23.98 **	5,140,835.70 **
Cultivar (Cul)	2	15.45 ^ns^	25.29 ^ns^	34.65 **	246.48 **	3,311,895.22 **
ID × Si	1	225.55 ^ns^	328.49^**^	5.85 *	2.02 ^ns^	1459.71 ^ns^
ID × Cul	2	48.63 ^ns^	10.21 ^ns^	1.86 ^ns^	0.60 ^ns^	58,972.55 ^ns^
Si × Cul	2	216.11 ^ns^	175.64 **	6.64 **	3.13 ^ns^	497,508.97 **
ID × Si × Cul	2	297.24 ^ns^	335.98 **	2.14 ^ns^	0.16 ^ns^	52,283.23 ^ns^
Error B	30	133.05	13.85	0.88	1.04	65,291.85
CV (A) (%)	-	15.85	8.44	22.57	16.57	20.22
CV (B) (%)	-	21.04	14.40	13.52	17.95	11.85

^ns^—not significant; **—significant at *p* ≤ 0.01; *—significant at *p* ≤ 0.05 according to F-test; DF—degrees of freedom; CV—coefficient of variation.

**Table 3 plants-14-01241-t003:** Irrigation depths applied at each phenological stage of cowpea cultivars.

Stages	Substages	Irrigation Depths (mm stage^−1^)	
		50% ETc	100% ETc
Vegetative	V0	7.79	15.57
	V1	4.61	9.22
	V2	9.57	19.13
	V3	11.84	23.68
	V4	9.98	19.97
	V5	14.14	28.28
	V6	18.11	36.22
	V7	17.53	35.07
	V8	17.98	35.95
	V9	12.54	25.08
Reproductive	R1	15.83	31.66
	R2	16.84	33.69
	R3	19.42	38.84
	R4	18.56	37.13
	R5	27.96	55.92
Total		222.69	445.39

## Data Availability

The dataset is available upon request from the authors. The data are not publicly available due to privacy restrictions.

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
