# Peer review of "Foliar Silicon Alleviates Water Deficit in Cowpea by Enhancing Nutrient Uptake, Proline Accumulation, and Antioxidant Activity"

_plants, 2025, doi:10.3390/plants14081241_

Round 1

Reviewer 1 Report

Comments and Suggestions for Authors

In this current version, presentation of result section is very unclear and must be significantly imporve to highlight major results of this study. Please modify it and consider major comments for a future evaluation of this work.

Reviewer 2 Report

Comments and Suggestions for Authors

In this manuscript, the effects of foliar silicon on cowpea under different irrigation conditions were studied. However, the experimental design was kind of simple and the demonstration of the results was not clear which makes this paper hard to read.

Comment 1: The introduction provides a thorough background on the importance of cowpea, the impact of water deficit on crop yield, and the potential benefits of silicon application. However, the connection between silicon and irrigation, especially the current knowledge about the influence of silicon on antioxidant system, nutrient uptake and accumulation was not well introduced.

Comment 2: Nutrient accumulation, osmotic substance content, antioxidant enzyme activity, growth index, water use efficiency and yield were measured, and the role of foliar silicon was evaluated. However, the specific molecular mechanisms of nutrient absorption, osmoregulation and antioxidant defense have not been explored in depth. Supplementing relevant molecular experiments, such as gene expression analysis and protein activity detection, will increase the depth of research by revealing the molecular mechanism of the role of foliar silicon.

Comment 3: The split-plot design, different irrigation depths, silicon treatments and cowpea varieties were set up. It would be better to have more detailed information about the cultivars used and their stress tolerance traits.

Comment 4: Soil silica fertilization is also a common fertilization method, and this manuscript does not explain why the treatment of foliar spray of silicon was chosen. In addition, the manuscript does not provide a clear introduction for the selection of 10 mg/L Si (OH)â‚„ as the silicon treatment concentration.

Comment 5: The figures and the related legends were not clear and well designed, which would make the results very hard to understand. In addition, the discussion should be reconstructed, the current version was too long, and the connection of current knowledge and the results of this study was not well discussed.

Comments on the Quality of English Language

Can be improved

Round 2

Reviewer 1 Report

Comments and Suggestions for Authors

All answers provide by authors are convincing.
